# Factors that Influence Farmers’ Views on Farm Animal Welfare: A Semi-Systematic Review and Thematic Analysis

**DOI:** 10.3390/ani10091524

**Published:** 2020-08-28

**Authors:** Agnese Balzani, Alison Hanlon

**Affiliations:** School of Veterinary Medicine, Veterinary Sciences Centre, University College Dublin, Belfield, D04 W6F6 Dublin 4, Ireland; alison.hanlon@ucd.ie

**Keywords:** farm animal welfare, farmers, perceptions, attitudes, empathy, human animal relationship, policy, stakeholders, communication, knowledge transfer

## Abstract

**Simple Summary:**

Farm animal welfare is a complex issue linking several academic disciplines, including animal science, veterinary, psychology, sociology, and agricultural economics. Farmers are one of the key contributors for the successful implementation of improved animal welfare standards. This review presents findings from the last 30 years into the factors that influence farmers’ views on farm animal welfare. Using findings from single and multidisciplinary studies, this review highlights the factors that influence the farmers’ views of animal welfare. Overall this literature review aimed to ask two main questions “what do farmers think (farmer’s general view) about farm animal welfare?” and “what are the factors that influence their thinking?”. This review demonstrates that a deeper understanding of how farmers view and value animal welfare can lead to more effective development of collaborative knowledge transfer, policies, and management initiatives directed at maintaining healthy animals. This work may serve as a checklist to implement further studies on stakeholder perspectives on animal welfare. It also provides recommendations on technical approaches and strategies to improve best practice on farm animal welfare.

**Abstract:**

Farm animal welfare (FAW) is a growing societal concern, reflected by over 30 years of research to inform policy and practice. Despite the wealth of evidence to improve FAW, implementation of good practice continues to be an issue. The role of the stakeholder, particularly farmers, is pivotal to FAW improvement. This semi-systematic review synthesizes the evidence published in the last 30 years, worldwide, to address two main questions “what do farmers think (farmer’s general view) about farm animal welfare?” and “what are the factors that influence their thinking?”. A thematic analysis was conducted to identify factors that influenced the implementation of FAW innovation. The main outcomes extracted from 96 peer-reviewed publications on a range of livestock species identified 11 internal factors including farmer knowledge, empathy, personality, values, and human-animal bond; 15 external factors including economic advantages, communication, time and labor influenced the perception of FAW. Farmers’ knowledge and cost implications of FAW were the most frequently reported factors. The review further highlights the need for promoting interdisciplinary collaboration and stakeholder participation. This study suggests strategies to improve FAW, including tools to support behavioral changes amongst farmers.

## 1. Introduction

Globally 70 billion animals are farmed annually for meat, milk and eggs. Two-thirds of these are farmed intensively [1]. In the growing debate about how food of animal origin is produced it is a challenge to disentangle the producer and stakeholder interests. Many factors contribute to the well-being and health of animals in commercial production systems including housing and environment; nutritional and health programs; handling and caretaker interactions; animal group dynamics; and common management practices. These factors have been established in more than four decades. 

However, despite the scientific progress in FAW, reflected by an annual publication growth of 13.3% [2], the implementation in practice is still poor in many areas. For example the level of neonatal lamb mortality has remained consistently high in some countries, despite 40 years of research to identify the risk and protective factors to reduce mortality [3]; another example is provided by Green et al. [4], which illustrates deficiencies in the implementation of ‘new’ best practice for the treatment of lameness in sheep.

As the world population continues to grow, the scientific community is facing a great challenge in order to sustainably increase agricultural production, to decrease food losses and maintain high animal health and welfare standards. Nevertheless, these efforts and innovations cannot be implemented without stakeholders’ support, including, farmers, veterinarians, agriculture advisors, consumers, policy makers, and retailers. 

Farmers are one of the key stakeholders for the successful implementation of enhanced FAW standards [5]. Since Seabrook’s [6] research on human and animal relationship (HAR), there has been a growing body of literature focused on exploring farmers’ attitudes towards animal health and welfare problems with the aim of optimizing the future programs designed to implement FAW. To better explore these aspects, socio-psychological approaches have been implemented since the late eighties [7]. Interdisciplinary studies focusing on farmers’ perceptions of FAW have reported that personalities [8], knowledge [9,10], values [11,12,13] economic advantages [9,13,14], communication with their veterinarian and agriculture advisors [10,15,16], time and management influence the perception of FAW. 

A deeper understanding of how farmers perceive and value FAW can lead to more effective development of extension programs, policies, and management initiatives aimed at maintaining healthy animals. Additionally, insights into factors that influence farmers’ decision-making process could support policy interventions aligned to the habitual behavior of target communities [17].

To this end, a review of the literature on factors which affect farmer decision-making regarding FAW is required to advance the approach to policy making and implementation in FAW. The aim of this review is to identify and understand the factors that can support behavioral change in farmers towards improved FAW, by synthesizing evidence collected from multiple disciplines focused on farmers’ perceptions that are relevant and have implications for FAW improvement. The objectives focused on two questions “what do farmers think (farmer’s general view) about farm animal welfare?” and “what are the factors that influence their thinking?”. The third objective was to provide recommendations on methodological approaches to exploring farmers’ perceptions to FAW, to serve as a link between the past and future research and help to inform and direct interdisciplinary research endeavors within this sphere.

## 2. Methodology

A semi-systematic review was conducted, this approach is intended for topics that have been conceptualized differently and studied by various groups of researchers within diverse disciplines and that preclude a full systematic review process [18]. The literature selection used the Preferred Reporting Items for Systematic Reviews and Meta-analyses (PRISMA) checklist (Figure 1). The inclusion criteria for papers were focused on farmers’ perceptions, opinions, values, beliefs, knowledge, and attitudes on FAW, were written in English and were published in a peer-reviewed journal. Geography and publication year were not restricted. The literature search was conducted from September 2019 to November 2019 and the databases used were PubMed, and Web of Science. Known relevant literature was used to develop search strings. The search terms had to be explicitly mentioned in the title, abstract, and/or in the keywords. The search terms were identical for both databases. All papers retrieved in PubMed were also recovered in the search using Web of Science. Figure 1 illustrates the selection steps and number of studies excluded at each step. The first step removed duplicate papers, the second step screened the title, abstract and keywords, and the studies that did not meet the inclusion criteria were excluded. Finally, the remaining articles were read and excluded if the research: (1) only presented animal welfare assessment results, (2) results did not meet the eligibility criteria, and (3) investigated effects of intervention aiming to change attitude.

The studies were classified and coded using NVivo 12 software (Nvivo 12, QSR international, London, UK). The review was structured in seven sections. Firstly, a descriptive analysis of the reviewed papers, including the years of publication, country where the research was conducted, the species, and data collection tools. Secondly the FAW topics of the reviewed studies are reported, including (i) farmers’ perceptions of animal welfare in general (i.e., values, HAR, personalities), (ii) and/or related to a species/system issues (i.e., aggression, painful procedures, housing, handling, naturalness), (iii) regulatory compliance, (iv) quality assurance programs, (v) economic and occupational aspects, (vi) and the implementation of innovation (i.e., anesthetics, enrichment). The third section describes the theoretical frameworks and methodological approaches used to assess farmers’ perceptions, attitudes, values, knowledge on FAW. The fourth section reports the outcomes of the thematic analysis of the studies reviewed that answered the questions: “what do farmers think (farmer’s general view) about farm animal welfare?” and “what are the factors that influence their thinking?”. The three last sections comprise the discussion, recommendations and conclusion of the semi-systematic review process.

## 3. Descriptive Characteristics of the Reviewed Literature

Ninety-six papers published from 1989 to 2019 worldwide were reviewed (Figure 2). The reviewed research came from Agriculture, Animal Welfare, Communication, Economics, Psychology, Sociology, and Veterinary Science disciplines. 

In addition, four reviews and four methodological papers were used as a guide to develop the search strategy. Two methodological studies developed, compared and validated new behavioral frameworks [19,20] for developing a scale to measure farmers’ attitudes to animal welfare and health. One modified existent methodology [21]. The last framework aimed to understand the variable characteristics and contradictory elements of farmers’ relationships with their animals [22]. More information about the methodological frameworks is provided in Table 2. All of the review papers were on health management of dairy cows and used different approaches. One of the review papers was a narrative literature review focused on cognitive processes involved in dairy farmers’ decision-making process related to herd health management [23]. A second review was a systematic review of studies on personality and attitude as risk factors for dairy cattle health, welfare, productivity, and farm management [8]. A third review was a narrative integrative style review that summarized studies focused on dairy farmers’ perceptions of lameness, claw health and the associated implications on the wellbeing and productivity of dairy cows [24]. The fourth review was a narrative review, focused on perspectives of farmers and veterinarians related to biological functioning (such as disease management), affective states (such as pain management), and concerns around natural living that have implications on the public’s acceptance of dairy farming [25].

Studies focused on farmers’ perceptions of FAW and opinions of FAW strategies have predominantly been conducted in dairy farming (*n* = 40) and pig production (*n* = 34) with other sectors less represented: beef (*n* = 12), broilers (*n* = 8), sheep (*n* = 5), layer hens (*n* = 7), goats (*n* = 1), turkeys (*n* = 1), donkeys (*n* = 1) (*n* = numbers of papers cited).

Data collection was mainly carried out via survey (*n* = 46). Interview was the second most common technique (*n* = 25). Other methods used to capture farmers’ perceptions of FAW were focus groups (*n* = 7), behavioral observation of farmers/farm workers and animals (*n* = 9), workshops (*n* = 4) and visual analog scale (*n* = 4). The statistical analysis used qualitative methods (*n* = 42), mixed methods (*n* = 34), and quantitative procedures (*n* = 19).

## 4. Animal Welfare Topics of the Studies

For the purpose of this review all the studies that investigated farmers’ perceptions, attitudes, values, knowledge of FAW were analyzed. Considering the scope of animal welfare science, the papers examined covered a wide variety of themes (Figure 3). Attitudes of farmers to painful procedures represented one of the main themes (pig tail docking *n* = 7; cattle disbudding/dehorning *n* = 10; claw trimming *n* = 1; pig euthanasia *n* = 1). Only 15% of the papers reviewed focused on understanding farmers’ perceptions and attitudes towards animal welfare in general. Analysis of data on farmers’ attitudes, beliefs, emotions and personality together with health management was the second most common theme investigated (cow and sheep lameness *n* = 8; mastitis *n* = 1; pig disease *n* = 2). Farmers’ attitudes towards participation in existing or improved quality assurance schemes was investigated in 15% of the studies reviewed.

## 5. Theoretical Frameworks and Methodological Approaches

In the early 1980s, small-scale research began to emerge studying the relationship between farmers and their animals [6]. It is only since Hemsworth et al. [7] that research of this nature has gained popularity. They developed a behavioral framework based on stockpersons’ opinion of pig behavior, the stockpersons’ own behavior toward pigs, and observations of the stockpersons’ behavior during interactions with pigs. The outcomes of their preliminary research supported the development of a training program for stockpersons to better understand animal behavior. Hemsworth et al. [26] developed cognitive-behavioral interventions that successfully targeted the key attitudes and behavior of stockpeople to support low-stress handling and thereby improve productivity. These insights enabled a deeper understanding of the approaches required to further assist farmers to improve the health and welfare of animals. To capture farmers’ attitudes scientists from many disciplines adopted a wide range of theoretical frameworks. Te Velde et al. [27] introduced the investigation of farmer (pig, dairy, poultry, and beef farmers) and consumer perceptions of the treatment of farm animals. Interviews were used based on a frame of reference (coping strategies) which consisted of values, norms, convictions, interests, and knowledge [27]. Many researchers in the agriculture and veterinary field mostly adopted the Theory of Reasoned Action and the Theory of Planned Behavior [28]. These theories propose that attitude, subjective norm, and perceived behavioral control are determined by an individual’s salient behavioral, normative and control beliefs, respectively. Attitude measures the extent to which an individual has positive or negative feelings towards the behavior in question. Subjective norm refers to the social pressures an individual may feel, perceived behavioral control refers to the anticipated ease or difficulty of performing the behavior in question [29]. This conceptual framework defines attitudes as being mediated through intention and as acting together with other explanatory factors, such as perceived control and subjective norm. In the reviewed literature, variations of this theory, other frameworks, and methodological approaches, were identified (Table 1).

The reviewed studies have been classified according to the methodological approaches used (Table 1), to facilitate the interpretation of the thematic analysis results and the comparison between studies.

## 6. Thematic Analysis of the Reviewed Literature

Thematic analysis resulted in 29 codes (Table 2), with three organizing themes which on analysis were determined to answer the questions: “what do farmers think (farmer’s general view) about farm animal welfare?” and “what are the factors that influence their thinking?” Farmers’ decisions to implement FAW innovation, could be explained by several reasons. Dessart et al. [17] reviewed the behavioral factors affecting the adoption of sustainable farming practices. They observed that farmers’ decisions to adopt FAW improvements were primarily related to business, occurred less frequently, often had long-term personal and economic consequences, may have involved large investments and long-term commitment (e.g., participating in voluntary land conservation programmes) and largely involved the provision of public good. In the reviewed literature a number of factors that influenced improvement in FAW have been identified using thematic analysis. Recurrent phraseology and topics identified in the reviewed literature were assigned to codes (AB), which in turn were classified into themes. Similar codes were grouped together for presentation in the results, whereas those that were seldom coded were only addressed in the discussion. In total 29 topics were coded. A comprehensive list of codes and the number of the times they were encountered in the literature is reported in Table 2.

### 6.1. Theme 1: Farmers’ Views of Animal Welfare

One of the purposes of this review was to address the question “what do farmers think (farmer’s general view) about FAW?” The interpretation of the concept of FAW differed amongst farmers with consistent patterns observed over thirty years, despite differences in methodologies, species, and topics. According to the three constructs developed by Fraser et al. [56], three farmer categories were identified according to their view on animal welfare. Good FAW for the majority of the farmers interviewed or surveyed aligned with satisfying the biological function of an animal (*n* = 25). Affective state of an animal emerged as the second most common view (*n* = 10). The third category related to the ability of an animal to engage in natural behavior (*n* = 8).

The main differences in perceived importance of each aspect of FAW are found in the ability of farmers to bond with their animals [48]. The relationship that farmers have with their animals depend on the species; life span; housing system, stocking density, and production system. Farmers that frequently handle their animals, for example milking of dairy cows, creates a high sense of attachment to their animals. Affective state and naturalness were considered good FAW in 55% of the studies focused on dairy farmers. Only 18% of the reviewed articles, which focused on pig farmers’ views of FAW referred to affective state being part of animal welfare. Enabling biological functioning was considered to satisfy the welfare needs of animals in 54% of the studies on pig farmers, 20% dairy, 10% broiler, 7% hen, and 6% of beef farmers ‘views on FAW (Figure 4). These results may be explained by the fact that the majority of the literature reviewed focused on dairy and pig farmers perception of FAW. Bock et al. [48] interviewed more then 400 poultry, cattle, and pig farmers in France, the Netherlands and Sweden, to ascertain factors that influenced HAR. This investigation showed that animal species does make a difference, even if one can encounter different levels of attachment with each species. Farmers generally felt closer to their cows than their pigs or chickens [48]. The lifespan of animals also influenced the bond between humans and animals. The attitudes, feelings and behaviors of farmers working with breeding stock tended to express varying degrees of emotional attachment whilst those preparing livestock for slaughter expressed varying degrees of emotional detachment [22]. Another common pattern found in the synthetized literature was that farmers considered all three concepts of FAW to be important (*n* = 30). Infrastructure was identified as a contributory factor to improve the HAR (*n* = 10). Available space, barn layout, housing conditions and equipment determined ease of management, which was associated with farmer well-being, better treatment of animals and, ultimately improved animal handling [36,57]. Organic systems have been noted to promote FAW and HAR [48,58].

Taking into account these findings from the reviewed literature, proximity with animals appears to be at the foundation of farmers recognition of the FAW. Systems that supported direct sensory contact between the farmer and their animals such as seeing, touching, speaking and listening were most likely to foster empathy and implementation of management strategies to improve FAW [36,48].

### 6.2. Theme 2: Internal Factors

A second purpose of this review was to address the question “what are the factors that influence farmers view on FAW?”. Research into the HAR, especially in the farming sector has a long history. Even if there is automation in livestock production, stockpersons are required to handle and regularly monitor animals. The level of care and interactions between stockpersons and their animals can influence productivity and welfare aspects [59]. Fear of humans has been negatively correlated with productivity in dairy, poultry and pig industries [26]. Numerous studies indicated that stockperson attitude towards animals influenced the behavioral response of animals to humans (fear of humans) and impeded their productivity. The results of the thematic analysis identified eleven internal factors that influence farmers’ attitude towards FAW.

#### 6.2.1. Knowledge

The majority of the farmers interviewed and surveyed in the reviewed literature referred to the importance of knowledge to influence their views on FAW. Farmers’ knowledge, skills and abilities were amongst the most important factors that influenced the implementation of FAW innovation [9]. In a review by Adler et al., [8] higher degrees of technical knowledge were reported to influence farmers’ perception of control and facilitate positive HAR. Furthermore, trained farmers [60] and farmers who understood the importance of their own actions [42] had lower risk of causing pain to livestock. Campler et al. [61] noticed that farmers that were clustered as confident and empathetic felt more confident and knowledgeable regarding identifying sick or compromised pigs compared with the unconfident and knowledge-lacking cluster.

Implementation of FAW depended on knowledge of FAW best practice. For example, 42% of the sheep farmers surveyed in UK did not know about the code of practice relating to the treatment of lame sheep [42,62]. Evidence suggests that farmers consistently underestimated FAW issues across livestock sectors and countries, this is exemplified by the underestimation of lameness in dairy cattle [8]. In the context of pig production, farmers’ perception of aggression in growing pigs and their opinion about mitigation strategies to reduce the expression of this behavior showed that some farmers in Germany were unaware that provision of enrichment is a requirement of EU Council Directive 2008/120/EC to control tail biting [13]. Early assessment of disease prevalence in farm animals and knowledge of the nomenclature was identified as a key opportunity to improve FAW and health in all species and production systems [63].

Research has demonstrated the influence of farmer knowledge and awareness on FAW improvement for decades e.g., [63]. The ability of farm advisors in delivering FAW knowledge contributes to the willingness of farmers to acquire such information [64]. Bassi et al. [36] conducted in-depth interviews to understand the factors that influenced routine practice in cattle farms (‘social practice’) and reported that farmers referred to the importance of family tradition, community, veterinarians, and industry experts in establishing the knowledge needed to carry out painful procedures.

The reviewed studies highlighted the importance of knowledgeable stakeholders regarding identifying sick animals and FAW best practices for the improvement of FAW.

#### 6.2.2. Empathy

Empathy has been shown to underpin positive management practices through the ability to anticipate the animals’ needs and to influence farmers’ views on FAW. Empathic responses towards animals are driven by a combination of experience and knowing individual animals, leading to the development of an understanding of the behavioral and cognitive similarities between animal and human experience (cattle, [65]). Kılıç et al. [44] surveyed sheep farmers about their perception of animal welfare and reported that all the participants believed that sheep are ‘sentient creatures’ and named their sheep, which also suggests that they have positive perceptions of animal welfare. Considering livestock as individual characters was suggested to reflect empathy with animals [66].

Language often used during interviews of UK cattle farmers suggested a sense of empathy and understanding that lameness could result in severe pain for the animals, for example: *“it is not very comfortable for the animal… or it is not very nice to see a cow hopping about”* [16]. Farmers often deliberately avoided becoming attached to animals with relatively short production cycles, destined for slaughter, such as fattening pigs and broilers to avoid attachment feelings [48]. In this regard many farmers expressed an unease about disbudding or dehorning because the operation inflicted pain on the animal [57]. Serpell [67] described this unease as an expression of the ambivalence between “affect” and “utility”. Wilkie [22] referred to this as the “productive paradox”.

In addition to anticipating animals’ needs and preventing problems, some authors studied the impact of animal-directed empathy on FAW and production. Kauppinen et al. [21] reported that empathy had no influence on either dairy welfare indicators or production, and comparable results have been reported for pig and cattle farmers. However, Kielland et al. [68] measured farmer empathy using a visual analogue scale, previously validated in human’s empathy studies, and found that cows owned by farmers that answered “no” to the statement “*animals experience physical pain as humans do*” had more skin lesions. In addition, dairy farmers who regarded their cows as intelligent beings, capable of experiencing emotions, knew and named their individual animals had higher milk yields [69]. Personality tests revealed that higher levels of empathy and job satisfaction were also related to higher milk yields [47].

From the reviewed studies empathetic indicators contributed to understanding how farmers view and make decisions toward FAW and can have a positive impact on production.

#### 6.2.3. Gender, Age, Years of Experience

The third most reported factor that influenced how farmers viewed FAW is demographic characteristic. The reviewed literature reported gender, namely females, to have a stronger perception of animals’ needs in pig [61] and sheep production [44]. Empathy related questions positively correlated with gender [61]. In pig farming, female producers on average gave higher scores than male counterparts for human emotional response, judgment of aggression severity and recognize fatigue in the animals [70]. Furthermore, in the same study, older farmers expressed greater motivation to intervene in pig fights than younger farmers. Age also influenced judgments of aggression score, younger (from 20 to 35 years old) participants had lower scores than older ones [70]. The same pattern has been observed in dairy and beef farmers. Older farmers with personal experience of cattle diseases had higher empathy scores [70] than younger producers (<39 years old) [53]. Knowledge of diseases and the complexity of treating the animals may be one factor making older producers more sensitive to animal pain [71].

Experience has also been reported to influence improvement of FAW. Families with a long history of farming had improved FAW practices, such as being substantially less likely to use an electric prod and herding tools on cattle ranches in California [60]. More experienced and older producers with smaller herds were keener to keep calves horned, as it was the traditional way of managing cows in small herds. Whereas, less-experienced and younger producers with larger herds considered disbudding to be a modern, and safer, way of managing dairy cows [72]. An open-answer question (“what do you think about tail docking”) was used to investigate the perception of tail docking piglets, showed that farmers with experience of non-docking, encountered less problems with tail biting compared to inexperienced farmers [73]. Furthermore, Kauppinen et al. [74] found no gender effect with FAW programme participation. Hence, it could be hypothesized that intentions to make improvements is possibly more determined by personal characteristics [51].

Taking into account these findings from the reviewed literature, female and experienced farmers were more empathetic with their animals and had a greater propensity to alleviate and minimize painful procedure.

#### 6.2.4. Social Norm-Pressure

The impressions that others have about farmers, and the farmers’ social context has been observed to influence farmers’ view and decision-making on FAW. The effect of subjective norm on intention to implement FAW suggested that farmers were influenced by the opinion of others [43,70]. Hansson and Lagerkvist [75] indicated that farmers’ decisions related to FAW were influenced especially by consumers and animal welfare inspectors. The authors stated that farmers were willing to improve FAW to gain the appreciation of other stakeholders and therefore improve profitability. Farmers in quality assurance schemes felt pressured by the demands of their buyers [76]. The same trend was observed in a small minority of Flemish broiler producers, who stated that there were few advantages in paying more attention to broiler welfare to improve their public image and gain consumer acceptance [77].

The Theory of Planned Behavior applied to study the attitudes of Finnish dairy farmers towards improving FAW showed that the agricultural advisors as a subjective norm was directly linked with milk production: elevated importance of the agricultural advisors, as perceived by the farmer, corresponded to lower mean milk production on the farm [74]. The authors stated that if a farmer draws on others’ opinions, he/she may feel insecure about taking care of his/her animals and therefore be more inclined to rely on the opinions and decisions of authorities like advisors, and veterinarians [78].

Results of the subjective norms model analysis also indicated that “family” and “neighbor farmers” influenced the opinion of dairy farmers. [46]. Furthermore, dairy farmers reported social pressure from known peers, although other farmers and veterinarians were considered to exert the greatest social influence. [53]. In contrast in the broiler sector, Flemish producers reported a lack of influence by peers, instead expressing a belief that they implemented higher standards of FAW than their neighboring producers [77].

There has been an increased societal pressure to reduce antibiotic use and improve FAW practices. In this regard social and advisory network approval, were the main factors influencing dairy farmers’ willingness to reduce antibiotic use [54]. Societal pressure was also indicated as a key driver for change in the pig sector in the context of tail docking of piglets [79].

Social desirability bias influences farmers’ responses and decision-making process, and in particular social norm pressure promoted the improvement of FAW.

#### 6.2.5. Non-Use Value

Farmers’ approaches to FAW are not just a question of rational and economic choice, attitudes and ethical views should be taken into account to understand their decision-making process [76]. Swedish researchers applied a “use values” and “non-use values” framework to understand the role of farmers’ personal evaluations on FAW achievement [11,38,39,75,80]. Non-use values referred to the economic value farmers derived from the welfare of the animals (i.e., economic value not derived from the direct use of the animal) [11,38]. They showed that non-use values in FAW were important motivational factors underlying dairy farmers’ decision making. For the first time a behavioral perspective was adopted in measuring the motivational factors for dairy farmers to improve FAW. From the questionnaire it emerged that farmers motivated to apply FAW standards were aware that they were treating their animals appropriately and not only for profit [11]. The element “animals feel good” was mentioned 228 times and thus was by far the most non-use value mentioned, other non-use-value recorded were: “avoidance of suffering”, “continue business”, “ethics”, “doing the right thing”, “animals eating properly” and “work environment” [75]. In addition, the concepts of motivation of farmers such as “instrumental business orientated” [21] or “intrinsic welfare orientated” [58] have been largely verified. Fischer et al. [81] to better understand the reasons behind certain practices and to deal with antibiotic resistance in agriculture showed that the emotional attachment that the farmers have to their cows and their sense of responsibility for them was central to their farm management. Other studies have similarly found how farmers applying health and welfare plans adopted significant measures to meet what they considered to be animal health. Sixty-eight percent of German beef cattle farmers interviewed expected personal and “non-monetary” benefits from their practices [82]. Bock et al. [48] provided a deeper understanding of the factors that influence HAR and showed that Brazilian farmers in all sectors (beef, dairy, pig, and poultry) felt responsible for taking good care of their animals and they perceived good farming as a key element of their job satisfaction. Borges et al. [50] identified the factors impacting Brazilian pig farmers to adopt FAW practices and noticed that behavioral belief was to “decrease animals’ stress”.

Moreover, evidence showed that farmers in specific animal welfare or organic quality assurance schemes were mainly motivated by ethical concerns and the possibility of improving FAW [48]. Similarly, whilst the interests of conventional pig farmers were mainly in economics, organic pig farmers were additionally interested in aspects related to animals, human health and the environment [83]. Farmers operating in the specific animal welfare and organic schemes did so primarily for ideological motives, believing that this production method granted pigs a better life [76].

From the results of this synthesis, farmers’ views of FAW can be better understood when taking into account the economic constraints and market incentives together with the production ethics and farmer morale.

#### 6.2.6. Dissonance Reduction

Applied cognitive behavior methodologies have been used to understand the factors that influence farmers’ views on FAW [27]. In-depth interviews showed a cognitive dissonance between perception (what a person says) and daily practices (what a person does), farmers’ perceptions of animal husbandry were based on a collective tradition with shared convictions, values, norms, and interests, and on knowledge that was derived from comparable upbringing, schooling, and daily experience on the farm, “The way things are” [27,36].

Bracke et al. [79], investigated the attitude of Dutch pig farmers towards tail docking, pointed out that conventional pig farmers’ views on tail docking, tail biting, and enrichment may arise from dissonance reduction. The authors’ observations showed that unwanted information was played down (e.g., about the painfulness of docking tails and the value of enrichment materials and curly tails), whereas advantages of tail docking were emphasized (tail docking is necessary). Kling-Eveillard et al. [57] conducted focus groups to understand dehorning perception; the study reported that farmers demonstrated cognitive dissonance that whilst they acknowledged concern for their animals, they believed that the painful practice was necessary.

Investigation of broiler producers showed ambivalence, namely the denial of FAW problems in broilers under their care, contradicting the negative view on broiler welfare in intensive production systems by the general public and authoritative scientific reports [77]. Cattle farmers have also been reported to underestimate welfare problems such as lameness among their stock [84]. Vanhonacker et al. [85] investigated citizens’ and farmers’ perception of FAW, reported a conflict in the farmers’ answers between their interest and values.

These reviewed studies demonstrated the instrumental relationships between individual farmers’ values, behavior and perception of animals’ needs. This knowledge is pivotal to stimulate and qualify the farmer’s decision-making in a way that will increase the farmers satisfaction and subjective well-being.

#### 6.2.7. Trust

The concept of trust in implementing FAW practice has been reported by several authors. Trusted advisors were very important to farmers’ receptivity to FAW advice [64]. For example, farmers perceived a trust-based dialog with their veterinarian, in particular when the latter trusted their farming competence [81,86].

#### 6.2.8. Personality

Since Seabrook [6] highlighted the link between farmers’ personality and animal behavior, many other studies have followed. Recently Adler et al. [8] reviewed the impact of stockpersons’ personalities and attitudes on dairy cattle welfare and found that cows displayed less abnormal behavior and more approach behavior to farmers who had a positive attitude. O’Kane et al. [42] investigated the association between sheep farmers’ personality and their barriers to improve FAW and reported that conscientiousness was associated with lower prevalence of lameness. Agreeableness and conscientiousness were farmer characteristics associated with their attitudes towards working with dairy cows [47]. Optimism has been reported to be a farmer characteristic positively related with work performance. For example, the degree of optimism may determine a farmer’s willingness to participate in FAW assurance schemes and voluntary disease control programs [87]. Self-assured and open-minded farmers were more likely to seek information about FAW and to hold principled views on FAW [78]. Empathy was positively correlated with the personality traits of extroversion, agreeableness, conscientiousness and intellect. Farmers who were more extroverted, conscientious and considered themselves intellectual were also more empathetic and had an indirect positive impact on FAW [47]. Borges et al. [50] surveyed 185 pig farmers to identify the beliefs underlying their intention to adopt environmental enrichment. They reported that the intention was mainly determined by their positive perceptions about their own self-identity [81].

Taking into account these findings from the reviewed literature, farmers’ personalities such as agreeableness and conscientiousness, positive and optimistic attitudes seem to be important internal factors that influenced how they viewed and implemented FAW improvements.

### 6.3. Theme 3: External Factors

#### 6.3.1. Costs

The economic disadvantages of implementing FAW emerged as a major influencing factor [9,13,14,48,78]. European farmers have criticized certain FAW regulations and measures for not being useful, detrimental to animals as well as difficult and costly to implement [48].

Different motivational orientations may explain the high heterogeneity among farmers in taking action to improve FAW. Regarding general attitudes towards animal welfare, Hansson and Lagerkvist [19], found that farmers gave the most importance to non-use values of FAW when they have an animal centered attitude. Other studies observed a similar trend, however, to be willing to implement changes in management and housing, or to increase the workload and even investment to improve FAW, farmers stressed the importance of a financial return [82]. Furthermore, financial considerations were key determinants for the FAW decision-making process, although this created emotional distress for farmers.

Overall, the review demonstrated that financial incentives were crucial for FAW improvement to optimize the economic performance, farmers’ mental well-being and continuity of farming activities.

#### 6.3.2. Herd Size—Management

Herd size and management have been extensively mentioned in the reviewed literature as main factors that influenced farmer thinking on FAW. Wikman et al. [88], assessed perception of pain in cattle and found that small herd farmers were more attuned to animal pain caused by disease than producers with medium or large herds. Similarly, farmers and stockpersons of small, medium pig farms had better knowledge and experience with sick animals than those that worked on larger farms [61]. The HAR has been shown to be negatively correlated with the size of the farm [65] and the recognition of animal health status [81]. In addition, management and housing system influenced the ability to bond with the animal [48].

Interestingly, Bock and Huik [48] studying the attitude of European pig farmers on FAW showed that farmers joined FAW assurance schemes to escape from the pressure of farm expansion by earning more per animal. Some farmers welcomed the stability provided in terms of their relationships with buyers and working in a more co-operative and planned way.

Overall, large herd size affects both the farmer’s emotional attachment to the animal and the possibility of judging animal health status, jeopardizing the implementation of FAW innovations.

#### 6.3.3. Communication

There is increasing evidence about the influence of communication between scientists, veterinarians and farmers on farmers’ views of FAW. Vigors [89], stated that “*the words used to communicate farm animal welfare to non-specialists may be more important than knowledge of welfare itself”*. Hambleton and Gibson [90], interviewed 110 veterinarians and 116 farmers about painful procedures in cattle and pointed out that the veterinarian-farmer communication was poor. The authors stated that improvements were needed to refine veterinarians’ understanding of farmers’ priorities and guiding clients on methods to improve calf welfare. Other studies similarly found that poor communication between veterinarians and farmers potentially undermined attitudes towards FAW legislation [10,91].

In-depth qualitative interviews have been used with dairy farmers in the UK to explore how they talked about the practices and processes of lameness. What emerged was that the language used to communicate with farmers played a major role in their understanding of the issues, interpretation of the lameness and therefore implementation of preventive measure [16]. Horseman et al. [16] adopted qualitative methods (in depth interviews) to investigate how farmers perceived lameness identification and treatment. The research demonstrated that the underestimation of lameness could be linked to the use of language to describe symptoms that under emphasized pain, impairing a prompt treatment of less severely lame cows. The misuse of the correct nomenclature was also found using a video analog scale and in farmer interviews to identify lame sheep. Whilst sheep farmers were able to identify lame sheep and used a consistent assessment method, there was a lack of knowledge regarding lesion type [4]. In this context Kristensen and Jakobsen, [23] reviewed evidence to provide researchers and veterinarians with a fundamental understanding on how to motivate dairy farmers to change behavior and to adopt FAW practices.

Jansen et al. [92] pointed out that effective communication must be tailored to each farmer’s perception of FAW and their specific needs. In addition, communication that is sensitive to farmers’ values may also function to increase their trust, and acceptance of, FAW policy, such as mandatory regulations and product certification schemes [80].

The synthesis of the reviewed literature highlights the central role that trust-based and tailored communication with veterinarians and advisors played in influencing farmers’ views and decision-making on FAW.

#### 6.3.4. Time and Space

Other important factors that influenced FAW improvement is the space and workload required to address animals’ needs [84,93,94]. Several authors have reported that farmers were reluctant to applying evidence-based FAW recommendations when it had implications for labour, space and time investment, regardless of the production system [9,16,72,76,77,81,86,94,95,96,97,98]. In contrast FAW interventions that could be carried out quickly, in a time-efficient manner, were reported to be easily adopted by farmers [86].

Interviews of farmers that implemented FAW improvements showed that investing in infrastructure that helped with animal handling, led in turn to happier workers and better treatment of cows [16,65].

#### 6.3.5. National Legislation—Quality Assurance Scheme—Paperwork

In several studies farmers referred to the unfair imbalance between national legislation and legislation elsewhere. Farmers’ views on FAW were influenced by the mandatory inspections for compliance with FAW national and quality assurance scheme regulations and the associated administrative workload. For example, focus groups showed that dairy farmers were stressed by paperwork, such as standard operating procedures required by the quality assurance scheme [48,64,82]. Furthermore, the growing numbers of regulations made Danish farmers feel insecure [91].

Inspections to assure that farmers are complying with the regulations on FAW can be based on different measures and audited by veterinarians or inspectors depending on the country [37]. Finnish pig and cattle farmers surveyed about their perception toward FAW considered that the increased FAW inspections following membership of EU were unfair [98]. A more recent survey carried out by Väärikkälä et al. [10] confirmed the same attitude towards FAW inspections. One third of farmers in the survey felt that FAW inspections violated their legal rights mainly because it was performed without prior warning, they did not understand why they were being inspected, they were not informed about the appeals process and/or the inspection report was not explained to them. However, Finnish farmers recognized that inspections were important to identify non-compliance with the standards. A similar attitude has been observed in Dutch and Danish farmers from semi-structured interviews conducted after an inspection [79,91].

These findings indicate that European farmers’ views on FAW were affected by the heterogeneity of legislation. Increased administrative workload and inspection requirements to assess FAW impacted farmer well-being.

#### 6.3.6. Niche Market Barrier

Uncertainties around the demand for animal friendly products and doubts about the economic advantage of participating in premium FAW standards have been reported [48,77,85]. Farmers skepticism was influenced by the failure of free-range and organic pig schemes in Germany, Austria, and Italy where the market for these products collapsed and were undermined by retailers and manufacturers [48]. Lack of confidence in the benefit of improving FAW has been observed to be strongly associated with the belief that animal friendly products will always be a niche market and do not achieve large market penetration [13].

#### 6.3.7. Tradition

“*We have always done it in this way*” is a recurring statement by farmers who do not adopt FAW innovation. Evidence shows that farming practices are learned through tradition and communities of practice [36,82,88]. Research aimed at exploring the implications of traditions on farming practices has shown that farmers with the same cultural, social, political and economic context developed a shared understanding of what it is to be a good farmer [81]. For example, some practices such as cattle branding have been referred to as important to bringing together family and community in an effort to not only carry out the practice, but to carry out the tradition [36].

#### 6.3.8. One-Size Fit All

Farmers’ criticism of the EU FAW legislation, which applies minimum national standards across all EU Member states was reported in the research literature. The main criticism stemmed from a ‘one size fits all’ approach’ which does not take account of the diversity of farming practices and policies across the EU [23,42,73]. For example, some countries implement higher FAW standards incurring additional costs for farmers but yet they are treated the same as countries with lower national standards when marketing their products. In contrast other member states had difficulty with complying with EU minimum standards due to the lack of knowledge, or resources such as equipment and housing facilities. The same criticisms have been directed to quality assurance schemes, farmers preferred to have tailored assessment measurements [97]. The importance of individually tailored tail biting prevention for different farms was referred to by pig farmers [73]. One-size fits all farm consultancy approaches were found to be redundant, with farmers motivated by a tailored communication to support the implementation of FAW innovation [23].

## 7. Discussion

This semi-systematic review and thematic analysis aimed to identify and understand the research focused on farmers perception, attitude, value, knowledge on FAW. The objective of this overview was to synthesise the evidence published in the last thirty years, worldwide, to address two main questions “what do farmers think (farmer’s general view) about farm animal welfare?” and “what are the factors that influence their thinking?”.

Whilst the reviewed literature had a common aim at capturing how farmers perceive FAW, the disciplinary approaches influenced the research outcomes due to heterogeneity in data collection and analysis methods. Thus, caution is required when drawing conclusions regarding the attitude and perceptions that influence farmers’ views on FAW. Outcomes from surveys compared with interviews or focus group are less informative, accurate and can create bias [99]. Paper based surveys conducted during workshops or focus groups may create a group-think, with individual farmers influenced by other participants [100]. Moreover, the studies designed using quantitative methods to analyze interview outcomes, may lead to misinterpretation. Quantitative methods are not suited for personalities and attitudes investigations, considering that interviews are context-related, and contain many non-quantifiable elements, creating a potential for inherent bias [99].

Several authors have highlighted the possibility of sampling bias (i.e., more welfare-oriented farmers participated in the studies) when studying farmer perception of FAW. Sampling bias can occur for several reasons such as: the sample is non-randomized, in which not all individuals have an equal chance of being selected; or randomized but with low response rate and/or confined only to a location, and/or with a voluntary participation. Therefore, the agreements or disagreements between studies make it difficult to draw a general conclusion.

Finally, the theoretical framework and methodological approaches applied to explore farmers’ perception, attitude, value, knowledge on FAW, led to heterogeneity in the terminology. For example, perception and attitude were often used interchangeably and numerous statements such “positive attitude”, and “values”, appeared extensively in the literature, however a clear definition was not always given. Pickens [101] explained the difference between attitude and perception. Attitudes define how a situation is seen, as well as define how we behave toward the situation or object [101]. Perception is closely related to attitudes. Perception is the process by which organisms interpret and organize sensation to produce a meaningful experience of the world. In other words, when a person is confronted with a situation or stimuli she/he interprets the stimuli into something meaningful to him or her based on prior experiences [101]. It was clear from the literature that a standardization of terminology is needed in order to facilitate the interpretation of the results and enhance their impact on FAW implementation.

However, regardless of the methodological approach used to explore farmers’ perception, FAW was in general considered to be important, even if only a small number of farmers recognized the need for improvement in FAW standards [12]. From the analysis farmers’ emotions and the HAR were important in the decision-making process. Findings showed that intensive farming systems provided the economic security for farmers rather than supporting their true ethical principles. Farmers were vulnerable to economic pressures that led them to take short-term decisions that might be contrary to their animals’ needs; thus, increasing farmer stress due to frustration. Several authors emphasized that achieving high productivity and giving animals a good life were in conflict [22,77]. Farmers have to manage their emotional bond to their animals in ways that still make it possible for them to treat them as commodities. The cognitive dissonance strategy represented a common example of how farmers might embody the tension between care and production [27].

As noted in the analysis, an important element in the development of good farmer skills is knowledge. The ability to transfer FAW knowledge by scientists, veterinarians, and advisors can inform decision-making by farmers. Improved education corresponded to increased job satisfaction for farmers and had a positive effect on perception of FAW for sheep [44], pig [61,96,102] and cattle farmers [36]. Empathy was an important influential factor of farmers’ views on FAW. Empathy has been shown to underpin positive management practices through the ability to anticipate the animals’ needs and “just know” when there is a problem emerging with their animals. Empathy was empowered through contact with animals, that in turn influenced the ability of farmers to appreciate the natural behavior, assess the likely affective state of livestock and provide for the biological needs of their animals. In large industrially managed systems, however, the lack of direct contact between the farmer and individual animals can hinder the development of the HAR [65]. Alternative and small to medium scale farm owners spent more time with their animals, building a strong HAR and ability to recognize sick animals. However, this finding needs to be interpreted with caution since the analysis compared studies that used different definitions of empathy and most importantly the analytical methods differed.

The results of this synthesis demonstrates that farmers’ views of FAW are influenced by the economic constraints and market incentives (value use), personal ethics, and farmer morale (non-value use). Taken together, whilst the profitability of the business was important, providing for the animals’ needs were equally or even more important [75]. However, the studies conducted by Hansson and Lagerkvist [11,19,39] had several limitations e.g., the survey had a low response rate and it was conducted in Sweden where FAW legislation exceeds the minimum standards established by European Directive, which may be indicative of a greater societal awareness and concern for animals. Another study found that farmers in Austria and Germany expected an improved job satisfaction (reduced stress and workload) when their animals had better welfare [82]. In contrast, Te Velde et al. [27] reported that farmers feared that improved FAW would increase workload and therefore impede working life. The same trend was reported for cattle farmers, where farmers perceived best care for the animals conflicted with labour demands or farm finances [81].

Borges et al. [50] surveyed 185 pig farmers to identify the beliefs underlying their intention to adopt environmental enrichment on their farms. They used Partial-Least-Square Structural Equation Modelling (PLS-SEM) to identify the impact of attitude, subjective norms, perceived behavioral control. The intention of farmers was mainly determined by their positive perceptions about their own capability to implement environmental enrichment (perceived behavioral control), followed by their perceptions about the social pressure to adopt it (subjective norms), their positive evaluations of adoption (attitude), and self-identity.

The effects of social desirability bias interfered with farmers’ responses and decision-making process; social norm pressure greatly influenced the implementation of FAW. A recent review of FAW policies in Germany, France, Italy and UK showed that farmers were greatly influenced by the level of societal concern [103]. Social norm is part of the TPB and therefore has been extensively investigated using this framework. In the TPB, people are assumed to include subjective norms in their conscious deliberations as to whether or not to perform certain behavior. Research on descriptive norms, on the other hand, has shown that they largely operate outside people’s awareness and that people are more inclined to conform to the behavior of similar others than to that of dissimilar others [104]. In this regard, several authors underlined the possibility of bias when studying the influence of social norm on farmer perceptions of FAW. Farmers denied that other stakeholders influenced their behavior, however, in-depth interviews revealed that farmers were more affected by stakeholders like the veterinarian, the advisor or the bank than they thought [17,53,55]. In this regard, individuals tend to deny other peoples’ influences on their own actions [105], which suggests that people are generally unaware of the influence that social norms have on them [52]. Moreover, scientists exploring the association between animal and farmer well-being showed that social networks provided a source of both social and professional support. Lack of support in their daily work was a catalyst for strain and was associated with farmer perception that their work was less acceptable, indirectly influencing FAW [87].

Another important aspect that emerged from the analysis was the considerable differences between countries in FAW policy and markets. These can be explained by political traditions, systems and cultures and/or differences in policy or regulatory styles [103]. Interestingly farmers’ perceived FAW inspections more negatively according to the communication skills of the inspector [91]. In this regard it has been questioned if the inspector should motivate farmers to promote respect for animals or continue to only check for compliance with FAW regulations [91]. Bock and van Huick [48] reported that farmers perceived differences between national legislation and legislation elsewhere to be unfair. It has also emerged that in Finland farmers would like to be more involved in the regulation making process and they wanted more uniform standards across all EU countries [10]. In this regard farmers often complained that their knowledge and experience was not acknowledged or used when FAW policy is made [106].

In the last decade EU legislative strategy for FAW has shifted from regulatory to voluntary approaches in cooperation with the private sector [107]. This shift has opened market opportunities for higher standards of FAW through quality assurance programs, although there is concern that marketing claims are not supported by a scientific evidence-base, creating inconsistent messages that are at odds with consumers’ needs for improved traceability and correct labelling [108]. Undoubtedly, the EU cross-compliance model and its lack of transparency is detrimental for FAW implementation [108].

Quality assurance programs can give farmers an opportunity to be recognized for their stewardship of FAW by their local community and consumers alike [108]. However, the lack of clarity and transparency of private standards in FAW can pose a risk for the credibility of the farmers [107,108]. The most important barrier to participating in FAW schemes was farmers’ distrust in the economic advantages of doing so, and some farmers believed that participation in these schemes would increase their economic risk. Lack of confidence in the benefit of improving FAW was associated with the belief that animal friendly products are a niche market and do not achieve large market penetration [13]. They doubted consumers’ willingness to pay and had little belief in the economic viability of such schemes [48]. On the other hand, a study to explore the attitude of European pig farmers towards FAW assurance scheme, reported that some farmers were attracted by the opportunity for high quality production and better labor conditions [48].

## 8. Recommendations

The reviewed literature and thematic analysis have highlighted the strategic approaches required to further assist farmers and other industry players to adopt additional measures to safeguard FAW. Critical success factors include the HAR relating to herd or flock size; knowledge transfer within farming and policy development of FAW in partnership with ‘actors’ such as farmers.

### 8.1. Human-Animal Relationship

The proximity of farmers and stockpersons to livestock and the frequency of interactions influences the HAR, empathy and other internal traits important to FAW. Whilst factors such as duration of the production cycle can impact the HAR and may be more difficult to address, issues such as the stockperson to animal ratio are likely to play an important role in the development of a positive HAR [109]. Developing guidance on staffing ratios, tailored for production systems and cycles within systems such as calving and lambing, when there are additional animals to care for, will help to support the HAR.

### 8.2. Knowledge and Knowledge Transfer

Creating a culture of continuous learning that is authentic and tailored towards farmers’ needs will support knowledge of FAW and underpin changes in attitude and behavior towards enhanced FAW practices. Peer to peer learning provides authenticity and may help to shape the social norms within the farming community. Horseman et al. [16] suggested that farmers sharing their positive experiences, for example of modifying their handling facilities, can be an efficient way to encourage other farmers to make similar changes.

The research on knowledge transfer of FAW indicates that the role of veterinarians and farm advisors is important to provide a trusted source of information. Enthusiasm, commitment and knowledge about the production systems, and a genuine interest in understanding and stimulating the farmer’s decision-making process from the farmer’s perspective, were key elements to understand farmers’ goals and better tailor communication [23]. This knowledge can be used to target policies according to farmer heterogeneity, i.e., their personality, degree of resistance to change, risk tolerance, level of moral and environmental concern and farming objectives [17]. Dessart et al. [17], suggested to improve knowledge by raising farmers’ awareness of FAW innovation, for example through the extension of advisory services. In this regard the use of digital technology has been successfully implemented [21,70,110], providing access and knowledge of information sources, including professional magazines, government advice, farmers’ organizations, feeding companies or slaughterhouses [48]. Bassi et al. [36] investigated the use of routine practices in cattle showed that drivers of change were often explained through changes in the accessibility of materials, competences, and meanings (the meaningfulness associated with a less-stressful practice). The use of multimedia tools to provide cognitive behavioral interventions showed positive results for changing dairy farmers practices [26] and pig farmers [70]. Dahl-Pedersen et al. [110] advocated the beneficial impact of training different professional groups about dairy cow welfare during transportation. Moreover, improving the knowledge of stockpersons has been shown to be beneficial for animal-stockperson relations such as ease of handling and reduced stress indicators in livestock [59]. These experiences offer a base to encourage farmer training on how to better assess the impact of poor welfare.

### 8.3. Policy Development

Developing FAW policy that is practical and implementable and that involves stakeholders and actors in the process will support greater compliance with FAW standards. The range of tools for designing and evaluating EU (agricultural) policies is broadening to include behavioral tools e.g., the European Commission’s “Better Regulation Toolbox” [111]. However, rather than traditional top-down approaches, creating integrated and diversified policy programs involving all stakeholders including farmers are required to facilitate the understanding and improvement of FAW [17,112]. It is in this context that “expert” and “non-expert” knowledge and understandings of, and concerns about, FAW contribute to the implementation of FAW innovation and the alternative modes of governance which they feed into [112].

Cost benefit analyses have been used successfully to motivate farmers, providing evidence about the economic losses due to poor welfare, and thus encouraging farmers to enhance FAW [9,37]. Benchmarking has been an effective strategy for improving calf welfare [113]. In fact, motivation has been shown to be a key factor to support behavioral change and implement practices to support FAW [80]. Furthermore, providing real time feedback to farmers through routine data capture for example from meat inspection can help farmers, veterinarians and advisors to inform herd health planning [102].

Finally, educating consumers and citizens about FAW is likely to underpin successful strategies to support farmers implementation of FAW innovation. Research has established that whilst there are commonalities and shared concerns about FAW amongst EU citizens and between non-experts and experts, there were also significant differences [114]. Addressing consumers’ concerns about FAW, increasing their awareness of agricultural practices and their willingness to buy more animal welfare friendly food will support FAW.

## 9. Conclusions

A semi-systematic review and thematic analysis was conducted to identify farmers individual characteristics and external factors that influence the implementation of FAW innovation at farm level. The evidence highlighted the instrumental relationships between societal and individual farmers’ values, personality, behavior and perception of animal’s needs. This knowledge is fundamental to stimulate and qualify the farmer’s decision-making in a way that will increase the farmer’s satisfaction and subjective well-being.

Farm animal welfare remains an important societal and economic concern. Nevertheless, behavioral changes among stakeholders are necessary for the realization of such a paradigm shift and adopting a shared responsibility, underpinned by improved communication.

Further research in this field should take into account the social network of livestock production in which the veterinarian, farmers, researchers, and advisors contribute to knowledge on FAW that is translated into on-farm application. Educating and involving stakeholders in the development of FAW innovation are key determinants of the success of such a system. More emphasis should be placed on tailoring solutions towards improving how stakeholders acknowledge the existence of the problem and their responsibility to act accordingly. For example, including farmers, consumers, and policy makers, in the FAW debate, accounting for their perception of the feasibility and cost effectiveness of any recommended management strategy could bring feasible innovation on FAW.

## Figures and Tables

**Figure 1 animals-10-01524-f001:**
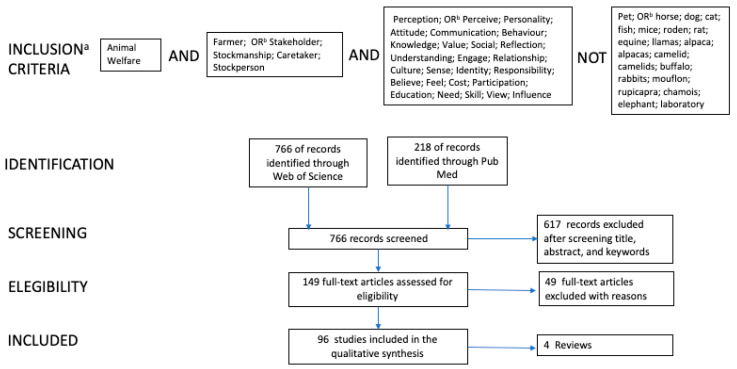
Number of publications captured in final database, searches, and removed at different stages of the paper selection process. ^a^ Inclusion criteria are linked with the Boolean AND, NOT operators. ^b^ For each criterion listed in the boxes the Boolean OR operator was used.

**Figure 2 animals-10-01524-f002:**
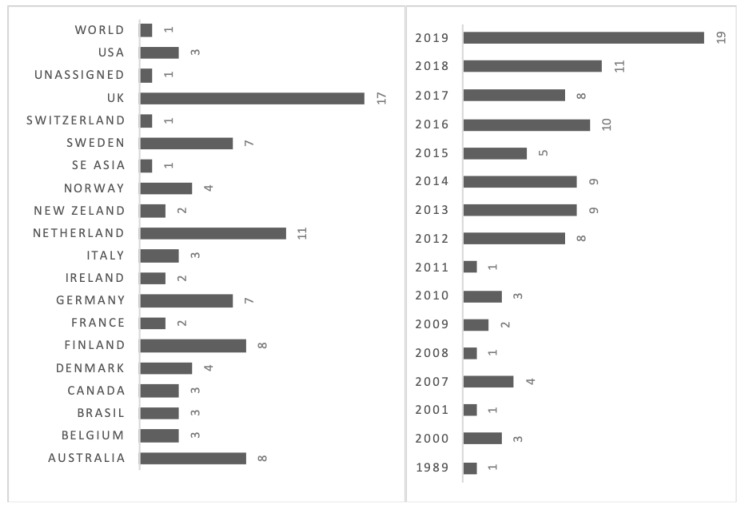
Number of reviewed articles (*n* = 96) focused on farmers’ perceptions, attitudes, values, knowledge of farm animal welfare (FAW) classified by countries and years of publication. Publications could have more than one country.

**Figure 3 animals-10-01524-f003:**
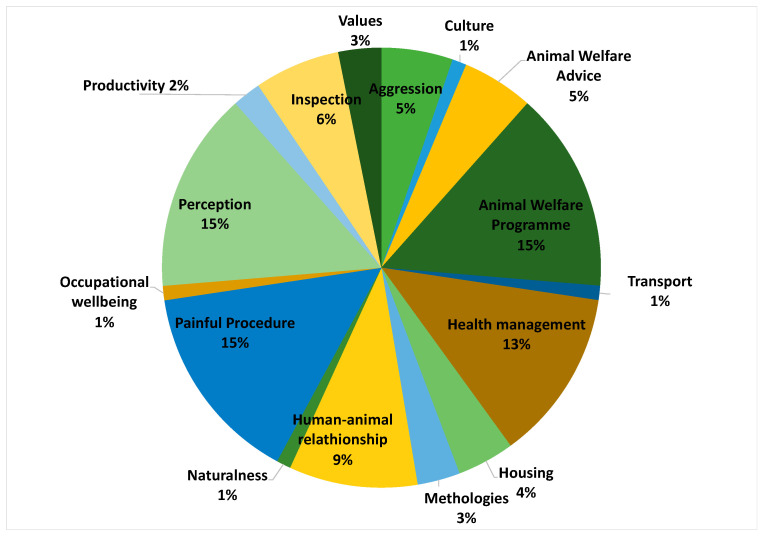
Range (%) of topics in the reviewed articles (*n* = 96) focused on farmers’ perceptions, attitudes, values, knowledge on farm animal welfare (FAW).

**Figure 4 animals-10-01524-f004:**
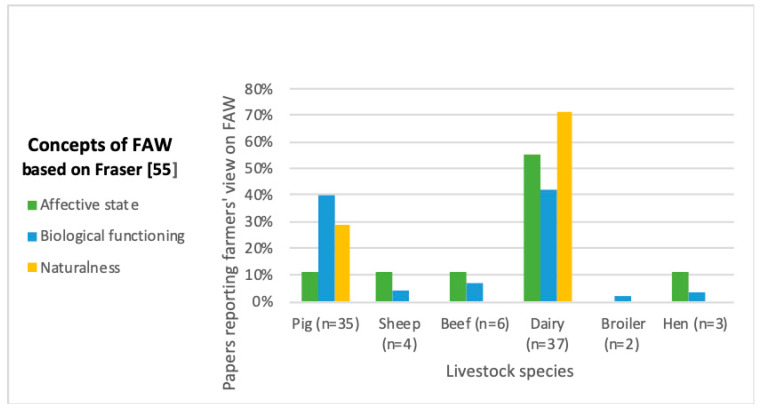
Percentage of publications reporting farmers’ (broilers, beef, dairy, hens, pigs, and sheep) views of farm animal welfare (FAW) based on Fraser’s Three Constructs [56].

**Table 1 animals-10-01524-t001:** Names, descriptions, authors, and references of the theoretical frameworks and methodological approaches applied to study farmers’ perceptions, attitudes, values, knowledge of farm animal welfare (FAW) in the reviewed articles.

Name	Description and Author
Co-design	Engagement among different stakeholders to identify similarities (and differences) in opinions to address and collaboratively design new practices in animal production [30,31,32]
Coping Strategies	Serpell [33] has stated that people, when using animals for certain purposes (milk, meat, and affection) always experience feelings of guilt as part of the human animal relationship (HAR) [27,34].
Critical Incident Methodology	The Critical Incident Technique is a set of procedures used for collecting direct observations of human behavior that have critical significance and meet methodically defined criteria [34].
Social practice theoretical framework (ethnographic data)	Shove et al. [35] emphasized that social reproduction and transformation stem from similar arenas and can be captured within the interconnected foundations of materials, competences, and meanings [36].
Framework of use and non-use values	McInerney [37] and Lagerkvist et al. [38] recognized that farmers may obtain economic or non-economic value from working with their livestock [39].
Interpersonal reaction inventory (IRI)	Davis [40,41] introduced the IRI Index that measures the dispositional empathy. Empathy consists of a set of separate but related constructs [42].
Illness Perceptions Questionnaire-Revised	Moss-Morris et al. [43] developed a questionnaire for human health, to investigate illness perceptions and emotional reactions [42].
Qualitative behavior assessment	Hemsworth et al. [7] developed cognitive-behavioral modification techniques based on the idea that people have a schema for a particular set of objects which can be used to retrain farmers’ behavior, as well as change their attitudes and beliefs [44].
Reflective model	Jarvis et al. [45] developed a measuring model and scaling technique to quantify attitudes. The reflective model assume that the attitudes are guided by the measured indicators [19].
Formative model	Jarvis et al. [45] represented the formative model when the measurement indicators are guided by the attitudes, implying that the attitude is defined by his indicators [19,21].
Discrete choice experiment (DCE) based in Social interaction theory	Becker [14] developed a method that determine and quantify factors that may influence choices. This method reveals determinants of farmers’ adoption behavior and derives the amount of monetary compensation necessary to encourage a choice [46].
The International Personality Item Pool (TIPI) personality test	The ‘big-five’ traits, which assessed extraversion, agreeableness, conscientiousness, emotional stability and intellect. The 50-item questionnaire was obtained from the International Personality Item Pool [42,47]
Wilkie’s Framework (ethnographic data)	Wilkie’s [22] analysis suggested that affection and attachment are dependent on the function of the animal and the phase at which an animal is in it’s production cycle. The framework evaluates HAR, where the relationship is varying in degrees of attachment and detachment. Four types of HAR (1) Concerned detachment, (2) Concerned attachment, (3) Attached attachment, (4) Detached detachment [48].
Theory of Planned Behavior (TPB) and Theory of Reasoned Action (TORA)	Ajzen [28,29] defined attitudes as being mediated through intention and as acting together with other explanatory factors, such as perceived control and subjective norm. They are designed to predict and explain human behavior in specific contexts (i.e., specific behaviors rather than aggregate behavior) [8,42,49,50,51,52,53,54,55].

**Table 2 animals-10-01524-t002:** Number of reviewed articles per code (*n* = 29 codes in total) identified using the thematic analysis. Publications could have more than one code. The codes are organized in three main themes, focused on farmers’ views (*n* = 3 codes) of farmer animal welfare (FAW), internal (*n* = 11 codes) and external (*n* = 15 codes) factors that influence FAW implementation.

Codes	Organizing Theme
**Biological functioning (23)**	Farmers’ views of FAW
**Naturalness (8)**	
**Affective state (8)**	
**Knowledge (28)**	Internal factors (farmers’ characteristics)
**Empathy (18)**	
**Gender, age, years of experience (13)**	
**Social norm-pressure (11)**	
**Interaction with animal (10)**	
**Non-use value (10)**	
**Education (8)**	
**Dissonance reduction (7)**	
**Trust (5)**	
**Personality (3)**	
**Optimism (1)**	
**Costs (28)**	External factors
**Herd size—management (21)**	
**Communication (11)**	
**Time (10)**	
**Labour condition (7)**	
**Legislation—paperwork (7)**	
**Space (7)**	
**Niche market barrier (6)**	
**Risk of disease (6)**	
**Tradition (6)**	
**FAW program (5)**	
**Best practice (4)**	
**Feasibility (4)**	
**One-size fit all (4)**	
**Feedback—slaughter (3)**	

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
