# Peer review of "Factors that Influence Farmers’ Views on Farm Animal Welfare: A Semi-Systematic Review and Thematic Analysis"

_animals, 2020, doi:10.3390/ani10091524_

Round 1

Reviewer 1 Report

This paper presents the results of a systematic review and thematic analysis addressing farmers’ attitudes and perceptions of farm animal welfare (FAW). It encompasses 96 papers to address how farmers think about FAW and the factors that influence their thinking. Farmers’ views of FAW account for biological functioning, naturalness and affective state. Internal factors that influence the implementation of FAW include knowledge, empathy, farmer gender/age/experience, social norm pressure, non-use value, dissonance reduction, trust, and personality. External factors that affect the implementation of FAW include costs, herd size and management, communication, time and space, legislation, niche market barrier, tradition, and the one-size fits all approachThe authors conclude by recommending improvements to the human-animal relationship, knowledge transfer, and policy development. 

Overall, this is an interesting review of farmers’ views of FAW and factors or barriers that may affect the implementation of FAW improvements. It is very thorough, with a sound methodology and a complete discussion of each area/factor, plus a thorough discussion of the limitationsI appreciated the summary at the end of the discussion of each factor, too. 

I believe the paper could benefit from a good review to correct some minor errors and inconsistencies. Examples (non-exhaustive list):  adding apostrophe to farmers in title, at line 36, line 66, and elsewhere where appropriate to signify possession; consistency of using or not using dash between semi and systematic; adding period to end of sentence at line 25, line 190, line 286 before 42%; consistency of verb tenses within lists in sentences, like changing decreasing” to “decrease” in lines 57-59; completing quotations in line 206, line 402; use of semi-colon vs. Comma in list line 234. 

My specific comments follow: 

  • Line 2: What does “Stage 1” mean? It is not referenced anywhere in the paper. 
  • Line 16: “presenting the state of the art of the human...” I do not understand what you are trying to say. Do you imply that the review presents the state of the art (in which case I believe you need to change the verb tense)? How do you qualify state of the art? I agree that it is quite thorough. 
  • Line 85: How does a semi-systematic review differ from a systematic review? Can you elaborate on these differences in the text? 
  • Line 114: You identified 766 records through Web of Science and 218 through PubMed, but only screened 766 records after removing duplicates. Were ALL the papers from PubMed duplicates of those from Web of Science? 
  • Lines 131+: Are these a list of the review papers on health management? This is unclear, because the sentences on their own seem like incomplete ideasYou could make this clearer by writing “One of the review papers was a narrative literature review...” etc., if this is in fact a list of the review papers. 
  • Line 148: Perhaps you mean “mixed”? 
  • Line 151: Do you mean “number”? 
  • Line 169: Do you mean knowledge on “farm” animal welfare? 
  • Line 222: Do you mean farmers views OF FAW? (missing of) 
  • Lines 227-31: You have essentially outlined the three circles approach by Fraser et al., without addressing the fact that these three elements of animal welfare come from an established framework. Do you think the farmers’ views match this framework because it is how animal welfare has been described to them? Is it a coincidence that their conception of FAW matches this framework? OR did you use this framework as the basis for your codes? 
  • Lines 290-294: Are you suggesting that environmental enrichment reduces aggression, and thus they should know about enrichment? The link between recognizing aggressive behaviour and not knowing about enrichment benefits is not clear here. 
  • Line 445: do you mean what a person “does”? 
  • Lines 468-471: It is interesting that trust emerged as a theme. Was there any aspect of the farmers’ holding the public/consumers’ trust? Perhaps farmers who believe that consumers are trusting them to make proper decisions with regards to animal welfare would be more likely to implement measures to improve animal welfare, because they hold themselves accountable to the public? Perhaps this was not covered in any of your reviewed papers. 
  • Line 480: Here you have included optimism within personality (which I believe is appropriate), but you have them as separate codes in Table 2. What is the reason for this? 
  • Line 564: Similar to the above comment, you discuss timelabour and space together, but have them as separate codes. Why did you choose to code them separately? 
  • There are a few codes in Table 2 that are not discussed or addressed (e.g. risk of disease, feedback-slaughter). Is there a reason why have you chosen not to discuss these? 

Author Response

Many thanks for your valuable edits and comments that have improved the quality of the manuscript.

My specific comments follow: 

Title. Apostrophe to farmers was added and stage 1 removed.

Throughout the text apostrophe to farmers was added where appropriate to signify possession

  • Line 2: What does “Stage 1” mean? It is not referenced anywhere in the paper. 

Stage 1 was mistakenly added as a requirement for manuscript submission. It has now been removed.

  • Line 16: “presenting the state of the art of the human...” I do not understand what you are trying to say. Do you imply that the review presents the state of the art (in which case I believe you need to change the verb tense)? How do you qualify state of the art? I agree that it is quite thorough. 

L 15.  text has been rephrased as follows:

Using findings from single and multidisciplinary studies, this review highlights the factors that influence the farmers’ views of animal welfare. Overall this literature review aimed to ask two main questions ‘what do farmers think (farmer’s general view) about farm animal welfare?’ and ‘what are the factors that influence their thinking?’

  • Line 25 adding period to end of sentence

L 22.  text has been rephrased as follows:

‘This work may serve as a checklist to implement further studies on stakeholder perspectives on animal welfare.’

  • Lines 57-59; change verb

L 60. replaced ‘decreasing’ with’ to decrease’

  • Line 85: How does a semi-systematic review differ from a systematic review? Can you elaborate on these differences in the text? 

L 89. replaced  ‘ an approach designed for topics that have been conceptualized differently and studied by various groups of researchers within diverse disciplines’ with the words ‘this approach is intended for topics that have been conceptualized differently and studied by various groups of researchers within diverse disciplines and that preclude a full systematic review process’

  • Line 114: You identified 766 records through Web of Science and 218 through PubMed, but only screened 766 records after removing duplicates. Were ALL the papers from PubMed duplicates of those from Web of Science? 

L 99. added ‘All papers retrieved in PubMed were also recovered in the search using Web of Science.’

  • Lines 131+: Are these a list of the review papers on health management? This is unclear, because the sentences on their own seem like incomplete ideas. You could make this clearer by writing “One of the review papers was a narrative literature review...” etc., if this is in fact a list of the review papers. 

L 151- 161. text has been rephrased as follows:

L ‘One of the review papers was a narrative literature review focused on cognitive processes involved in dairy farmers’ decision-making process related to herd health management. A second review was a systematic review of studies on personality and attitude as risk factors for dairy cattle health, welfare, productivity, and farm management. A third review was a narrative integrative style review that summarized studies focused on dairy farmers’ perceptions of lameness, claw health and the associated implications on the wellbeing and productivity of dairy cows. The fourth review was a narrative review, focused on perspectives of farmers and veterinarians related to biological functioning (such as disease management), affective states (such as pain management), and concerns around natural living that have implications on the public’s acceptance of dairy farming.’

  • Line 148: Perhaps you mean “mixed”? edit accepted
  • Line 151: Do you mean “number”? edit accepted
  • Line 169: Do you mean knowledge on “farm” animal welfare? edit accepted
  • Line 210: completing quotations edit accepted
  • Line 222: Do you mean farmers views OF FAW? (missing of) edit accepted
  • Lines 227-31: You have essentially outlined the three circles approach by Fraser et al., without addressing the fact that these three elements of animal welfare come from an established framework. Do you think the farmers’ views match this framework because it is how animal welfare has been described to them? Is it a coincidence that their conception of FAW matches this framework? OR did you use this framework as the basis for your codes? 

L 229. Text was rephrased as follow:

‘According to the three constructs developed by Fraser et al., 1997 three farmer categories were identified according to their view on animal welfare.’

  • Lines 290-294: Are you suggesting that environmental enrichment reduces aggression, and thus they should know about enrichment? The link between recognizing aggressive behaviour and not knowing about enrichment benefits is not clear here. 

L 294. the text has been rephrased as follows:

‘In the context of pig production, farmers’ perception of aggression in growing pigs and their opinion about mitigation strategies to reduce the expression of this behavior showed that some farmers in Germany were unaware that provision of enrichment is a requirement of EU Council Directive 2008/120/EC to control tail biting’

  • Line 402: completing quotations

the text has been rephrased as follows:

‘In this regard societal and advisory network approval, were the main factors influencing dairy farmers’ willingness to reduce antibiotic use.  Societal pressure was also indicated as a key driver for change in the pig sector in the context of tail docking of piglets.’

  • Line 445: do you mean what a person “does”? edit accepted
  • Lines 468-471: It is interesting that trust emerged as a theme. Was there any aspect of the farmers’ holding the public/consumers’ trust? Perhaps farmers who believe that consumers are trusting them to make proper decisions with regards to animal welfare would be more likely to implement measures to improve animal welfare, because they hold themselves accountable to the public? Perhaps this was not covered in any of your reviewed papers. 

L 470. Thanks for the comment. In the reviewed literature there is evidence that reports farmer perception of market and consumers regarding animal welfare and animal friendly products. For the purposes of this manuscript, we focused on farmer perception of animal welfare and did not discuss the relationship with consumer aspect.

  • Line 480: Here you have included optimism within personality (which I believe is appropriate), but you have them as separate codes in Table 2. What is the reason for this? See below
  • Line 564: Similar to the above comment, you discuss time, labour and space together, but have them as separate codes. Why did you choose to code them separately? See below
  • There are a few codes in Table 2 that are not discussed or addressed (e.g. risk of disease, feedback-slaughter). Is there a reason why have you chosen not to discuss these? 

Due to the length and complexity of the manuscript, the authors agreed to address the major codes (those most frequently reported in the reviewed literature) in the text, whilst presenting the full list of codes in the table. Similar codes were grouped together for presentation in the results, whereas those that were seldom coded were only addressed in the discussion. In line 216 additional information to guide the reader has been provided as follow:

 L214. Replaced ‘Two main organizing themes categorised as internal and external factors, comprised of eleven and fifteen sub-themes respectively (Table 2)’ with the words:

 ‘Recurrent phraseology and topics identified in the reviewed literature were assigned to codes (AB), which in turn were classified into themes.  Similar codes were grouped together for presentation in the results, whereas those that were seldom coded were only addressed in the discussion. In total 29 topics were coded. A comprehensive list of codes and the number of the times they were encountered in the literature is reported in Table 2.

Reviewer 2 Report

This work will contribute to the global body of work about farmers perceptions of Farm Animal Welfare and potential engagement with initiatives designed to improve it.   It was however a difficult text to follow.  Throughout the text used needed to be i. made more clear (a lot of knowledge seemed to be assumed) and at the same time ii. simplified. There were many instance of multiple verbs being used that were superfluous and the the meaning could be substantially improved by their meaning. The results section was difficult to digest following the really useful table.    I have made comments on the manuscript (supplied as a separate annotated document), mainly on the results section.  It was unclear where the SR and the TA started/ended.  I anticipate the once the writing is simplified and the content made clearer that the paper would be substantially improved in terms of publication prospects. 

Author Response

Many thanks for your valuable edits and comments that have improved the quality of the manuscript. Please see the attachment
